# Enhancing the Coherent Phonon Transport in SiGe Nanowires with Dense Si/Ge Interfaces

**DOI:** 10.3390/nano12244373

**Published:** 2022-12-08

**Authors:** Yajuan Cheng, Shiyun Xiong, Tao Zhang

**Affiliations:** 1School of Physics and Materials Science, Guangzhou University, Guangzhou 510006, China; 2Guangzhou Key Laboratory of Low-Dimensional Materials and Energy Storage Devices, School of Materials and Energy, Guangdong University of Technology, Guangzhou 510006, China; 3Laboratory for Integrated Micro Mechatronic Systems (LIMMS/CNRS-IIS), The University of Tokyo, Tokyo 153-8505, Japan

**Keywords:** core-shell superlattices, coherent phonon transport, thermal transport, molecular dynamics

## Abstract

The manipulation of phonon transport with coherent waves in solids is of fundamental interest and useful for thermal conductivity design. Based on equilibrium molecular dynamics simulations and lattice dynamics calculations, the thermal transport in SiGe superlattice nanowires with a tuned Si/Ge interface density was investigated by using the core-shell and phononic structures as the primary stacking layers. It was found that the thermal conductivity decreased with the increase of superlattice period lengths ($L_{p}$) when $L_{p}$ was larger than 4 nm. This is because introducing additional Si/Ge interfaces can enhance phonon scattering. However, when $L_{p}<4$ nm, the increased interface density could promote heat transfer. Phonon density-of-state analysis demonstrates that new modes between 10 and 14 THz are formed in structures with dense Si/Ge interfaces, which is a signature of coherent phonon transport as those modes do not belong to bulk Si or Ge. The density of the newly generated modes increases with the increase of interface density, leading to an enhanced coherent transport. Besides, with the increase of interface density, the energy distribution of the newly generated modes becomes more balanced on Si and Ge atoms, which also facilitates heat transfer. Our current work is not only helpful for understanding coherent phonon transport but also beneficial for the design of new materials with tunable thermal conductivity.

## 1. Introduction

The ability to accurately control the thermal transport of material is fundamental in the development of heat dissipation and energy conversion applications [1]. For waste heat energy harvesting with thermoelectric modules, materials with low phonon thermal conductivity (TC) are essentially required to maximize efficiency [2,3,4]. Among different TC reduction strategies [1,5,6,7], constructing a superlattice (SL) with different materials has been regarded as a promising approach. In SLs, phonons can be extensively scattered by the numerous interfaces. It is well known that the TC of a SL is period length-dependent, and it varies non-monotonically with period length. A minimum thermal conductivity (TC) was usually observed at a certain critical period length $L_{p}$ (typically a few nanometers) [8,9,10,11,12,13,14,15,16,17,18,19,20]. Beyond the critical period length, phonon–interface scattering is dominant, and TC increases with an increased period length. In this region, phonons can be treated as particles, and the Boltzmann transport equations can be applied to predict the TC of a SL [21]. However, phonon transport becomes coherent when $L_{p}$ is below the critical value, and the TC increases with a decrease of $L_{p}$. For SLs with short period lengths, the structure can be treated as a homogeneous material, and phonons will not feel the interface. With the decrease in period length, more and more phonons transport coherently, leading to an enhanced TC in SLs. The coherent phonon transport is a consequence of the wave effect, and the phonon particle assumption fails in this region. Coherent phonon transport has been widely observed in SLs with short period lengths [8,9,10,11,12,13,14,15,16,17].

The experimental observation of coherent phonon transport was realized by Luckyanova et al. [8]. By varying the number of periods in finite-thickness GaAs/AlAs SLs, they found that the measured TC was linearly proportional to the total SL thickness from 30 to 150 K, which is a signature of the coherent phonon conduction process. To demonstrate the transition from incoherent to coherent phonon transport, Ravichandran et al. [9] fabricated electrically insulating SrTiO${}_{3}$/CaTiO${}_{3}$ and SrTiO${}_{3}$/BaTiO${}_{3}$ SLs with atomically sharp interfaces. The measured TC versus period length clearly showed a minimum value when $L_{p}$ was around 2 nm at room temperature, indicating the transition from incoherent to coherent transport. Theoretically, Latour et al. [10] established a model to calculate the coherent length by taking the spatial correlation of atomic vibrations in molecular dynamics simulations. The results indicated that the coherent length in SiGe SLs was shorter than $L_{p}$ in the scattering region, while it became larger than $L_{p}$ in the coherent transport region. Except for the enhanced TC at short period lengths, the coherent phonon transport can lead to a vanished temperature jump around the interfaces if a temperature gradient is applied, which can be easily demonstrated by the non-equilibrium molecular dynamics simulations [11]. If phonon–interface scatterings exist, there should be a temperature jump around the interface. The vanished temperature jump around the interfaces is a direct consequence of coherent phonon transport as the entire structure can be treated as a homogeneous material in such a scenario and phonons do not feel the interfaces.

The coherent phonon transport offers the possibility to enhance thermal transport with more defects, which is extremely interesting in heat dissipation management. However, the current research has focused on the observation of the coherent transport phenomenon in different systems. Tuning the coherent strength by designing appropriate structures has not been demonstrated yet. Considering that the coherent phonon transport relies on short periods with homogeneous structures, increasing the interface density could be a promising strategy to further enhance the coherent transport in SLs with small period lengths. Actually, more interfaces can balance the atomic vibrations between two materials, i.e., reduce the vibration mismatch. As a result, the atomic vibrations in the entire structure become more homogeneous, and thus coherent transport could be enhanced.

In this work, we investigate the coherent transport phenomenon in nanowire (NW) SLs. Compared to the TC in bulk state, the TC of NWs is much smaller [5]. The appearance of soft flexural modes, phonon confinement, and phonon–surface scattering in NWs with small diameters make phonon transport more complicated. New phonon transport mechanisms such as phonon hydrodynamic transport might appear [22]. On the other hand, new engineering freedoms, such as transverse geometry modulation [7], orientation engineering, and surface engineering [23], can be applied to control the heat flow. As a result, it would be interesting to know whether the coherent phonon transport in SLNWs would be different from that of the corresponding bulk state or not. We combine SLs with core-shell and phononic structures to create lateral interfaces in SiGe SLs NWs. We name such combined structures core-shell SL (CSSL) and phononic SL. We try to clarify how those newly introduced interfaces affect thermal transport with different SL period lengths. We find that at a large period length, where the phonon particle effect dominates the transport, the additional interfaces will enhance the phonon scattering and will further reduce the TC. However, at short SL period lengths, the generated Si/Ge interfaces can promote coherent phonon transport by generating new phonon modes, which eventually enhances thermal transport.

## 2. Structures and Methods

We started with a normal Si NW with a diamond lattice along the [100] direction with a squared cross-section, the width of which was fixed as $L_{w}=4.4$ nm. Each layer of the SL was composed of a Si-Ge core-shell structure, as shown in Figure 1. Among the two primary stacking layers in SLs, one was composed of a Ge-core/Si-shell while the other one corresponded to a Si-core/Ge-shell with the same size. We denote the size of the core as $L_{c}$. The alternative stacking of the two primary layers (ABAB) formed a CSSL. When $L_{c}=0$ or $L_{c}=L_{w}$, the structure corresponds to a conventional Si/Ge SL. The period length $L_{p}$ is the total thickness of layers A and B (Figure 1). TCs are calculated for systems with different $L_{c}$ and $L_{p}$. It is worth noting that Si and Ge have slightly different lattice parameters (5.43 Å for Si and 5.64 Å for Ge). In real cases, there should be a lattice mismatch at Si/Ge interfaces. However, we will need to build up a huge simulation box to compromise the two lattices if the lattice mismatch is considered, which will increase the simulation load dramatically. On the other hand, considering the lattice mismatch at interfaces will mean that we are not able to investigate the lateral interfaces. This is because at short SL period lengths, it is not possible to build a mismatched lateral interface. As a result, we follow the strategy in most of SiGe SL simulations in the literature [16,18,24] by setting up coherent interfaces, i.e., interfaces without lattice mismatch. We use an averaged lattice parameter of Si and Ge for SLs. In such a consideration, small stress exists at the interfaces. We terminated our NW surfaces with (100) facets, which have been widely adopted for Si NW simulations [18,20]. During the simulations, the atoms on the surface of NWs would form new bonds to lower their energy, i.e., surface reconstruction. The full reconstruction of all surface atoms requires a long time for simulations. Therefore, to save the computational costs and avoid possible unreconstructed atoms, we manually introduced (2×1) type surface reconstructions, the structure of which is shown in Figure 1. The (2×1) reconstruction forms a series of dimers, decreasing the number of dangling bonds by a factor of two.

We performed equilibrium molecular dynamics simulations to calculate the TC of all structures. All simulations were done with the GPUMD package [25,26]. The Tersoff potential [27] was adopted to describe the interatomic interactions among different atoms. The time step was chosen as 1 fs, which was small enough to conserve energy during simulations. The velocity Verlet algorithm is used for the integration of Newton’s equation. For all systems, a length of 26.6 nm is used for TC simulations, which contains 24,432 atoms. We carefully tested that TC converges at a size (8.8 nm) much smaller than the length adopted (26.6 nm). All structures were first equilibrated in an isothermal–isobaric ensemble (NPT) at 300 K and 0 Pa for 5 ns to allow the relaxation of all atoms, especially the surface atoms. After that, the simulation was shifted to the canonical ensemble (NVT) for 1 ns and the microcanonical ensemble (NVE) for 2 ns to further relax the structures with different conditions. After the equilibrium, another 6 ns of simulation in the NVE ensemble was performed to collect the heat flux every 5 steps. The Berendsen thermostat and barostat [28] were used to control the temperature and pressure of the systems. The TC was then evaluated by the Green–Kubo equation [29]:(1)$$\kappa =\frac{1}{k_{B}VT^{2}}\int_{0}^{\infty }\langle J_{z}\left(\tau \right)\cdot J_{z}\left(0\right)\rangle d\tau$$
where $k_{B}$, *T*, and *V* are the Boltzmann constant, temperature, and volume of the system, respectively. $J_{z}$ represents the heat flux along the *z* direction (axial direction of NWs). $J_{z}\left(\tau \right)\cdot J_{z}\left(0\right)$ is the auto-correlation function of $J_{z}$, and τ is the correlation time. The maximum correlation time is chosen as 2 ns in our studies, which is one-third of the total heat flux production time. $\langle \cdots \rangle$ denotes ensemble averages. We only consider the heat flux along the axial direction *z* of NWs as the Green–Kubo method can only be applied to the direction with periodicity [30]. All TCs were averaged with 24 independent simulations starting with different initial velocities. The error bar is calculated as the standard error of the 24 independent simulations. The combination of equilibrium molecular dynamics simulations with Tersoff potential can provide reasonable results for TC of SiGe SLs [10].

## 3. Results and Discussion

Figure 2a shows that the TC of SiGe SL NWs and CSSL NWs with $L_{c}=2.2$ nm varies with the period length $L_{p}$ at 300 K. As a reference, we also calculated the TC of the pristine Si and Ge NWs with the same cross-section. The value was obtained as 58.1±5.2 and 25.0±1.8 W/mK for Si and Ge NWs, respectively. In nanostructures, the reduction of TC as compared to the corresponding bulk state arises from two aspects: phonon–surface scattering and phonon confinement. Although the phonon confinement effect is less important compared to the surface scattering effect in TC reduction, it is not negligible when the cross-section of NWs is small. The confinement can modify the phonon dispersion by flattening the dispersion curves close to the Γ point, thus reducing the phonon group velocities [31,32]. Since the cross-section size of our simulated NWs is only 4.4 nm, which is comparable to the wavelength of phonons near the Γ point, the confinement is expected to be non-negligible. Due to the perfect surface reconstruction, the TC of our pristine structures is larger than the unreconstructed ones with similar size [33,34]. The simulated TC of pristine Si and Ge is larger than the experimentally measured values with an even larger cross-section size [5]. Such a discrepancy should arise from the perfect surfaces adopted in simulations. For the NWs synthesized experimentally, the surfaces are normally covered by a layer with an amorphous structure, which can scatter phonons diffusively. In contrast, the perfect surfaces constructed in simulations are fairly speculative for phonons, especially for our reconstructed surfaces, where fewer dangling bonds are left compared to the unreconstructed ones.

For the normal SiGe SL NWs, there is a minimum TC around $L_{p}=4.4$ nm, which is the same as the corresponding SiGe bulk SLs [10]. The minimum TC in our SiGe SL NW is only 1.9±0.05 W/mK, which is 50-fold and 13-fold smaller than the value of the pristine Si and Ge NWs, respectively. The slope of TC with increasing $L_{p}$ is large and negative for $L_{p}<L_{p}^{m}$ ($L_{p}^{m}$ denotes the period length with minimum TC), and small and positive for $L_{p}>L_{p}^{m}$. The slope is much smaller compared to the one in bulk SiGe SLs [10] with $L_{p}>L_{p}^{m}$, which is ascribed to the phonon–surface scatterings. However, the slope is more negative than that in SiGe bulk SLs [10]. This phenomenon indicates that the coherent transport in NWs is more obvious. Since in large period length regions, phonons can be treated as particles, we may call this region the particle zone. In contrast, the coherent transport region corresponds to the wave zone.

With the introduction of core-shell structures in each layer of SLs (the core size $L_{c}$ is fixed as 2.2 nm), the TC is further reduced when $L_{p}>L_{p}^{m}$, which means the generated new interfaces by core-shell structures can further scatter phonons. As a result, the TC in SiGe CSSL NWs increases at an even lower speed with period length when compared to the SiGe SL NWs. In contrast, the TC in CSSL NWs is enhanced at short period lengths ($L_{p}<L_{p}^{m}$) compared to that of SL NWs. Since the coherent phonon transport dominates in such short period lengths, we believe the TC increase in CSSL is due to the enhanced coherent phonon transport.

To check how the core size affects the TC in CSSL NWs, we calculated the TC as a function of the core size $L_{c}$ (see Figure 1 for definition). The corresponding results are demonstrated in Figure 2b. Note that the width of the NW cross-section $L_{w}$ is fixed as 4.4 nm. Thus, the increase in core size will lead to the reduction of shell thickness. As a reference, the TCs of Si-core/Ge-shell and Ge-core/Si-shell NWs are also illustrated in Figure 2b. In general, the TC of Ge-core/Si-shell NWs decreases with the enlarge of Ge core size. In contrast, for Si-core/Ge-shell NWs, the TC reduces firstly and then increases with the enlargement ofthe Si core size, resulting in a TC minimum around $L_{c}=1.1$ nm. The increase of TC with the enlargement of the Si core could be ascribed to the larger TC of Si compared to that of Ge. For SiGe CSSL NWs with large period length ($L_{p}=26.6$ nm), the TC is almost unchanged as core size increases. However, in the coherent phonon transport region ($L_{p}=1.1$ nm), the TC of CSSL NW increases gradually with the enlargement of core size. Such an enhancement of TC should arise from the enlarged Si/Ge interfaces. The increased Si/Ge contact area due to the enlarged core size can make the vibration of Si and Ge atoms more balanced, which promotes coherent transport. To confirm the correctness of the equilibrium molecular dynamics method, we also performed homogeneous non-equilibrium molecular dynamics simulations [35,36] for CSSL NWs with $L_{p}=1.1$ nm. The corresponding results are illustrated by the open diamond symbols in Figure 2b. One could observe that the obtained TC is close to the results obtained by equilibrium molecular dynamics simulations. In particular, they possess a similar trend with the change in core size. The similar TC values obtained by the two independent simulation techniques confirm the validity of our simulations.

To illustrate the coherent transport in SL NWs, we calculated the vibrational density-of-state (VDOS) for the structures with different period lengths and core sizes. The VDOS of Si and Ge are separated to analyze the vibration frequency mismatch between them. The corresponding results are demonstrated in Figure 3. For the SiGe SL NW with a large period length ($L_{p}=13.3$ nm), the VDOS values of both Si and Ge atoms are close to the corresponding NW values. The VDOS vanishes beyond their Debye frequencies (16 THz for Si and 10 THz for Ge). With the reduction of period length, the VDOS of both Si and Ge changes: firstly, the position of the last peak for Si (∼15 THz) and Ge (∼9 THz) left and right shifts, respectively, resulting in a reduced distance between the two peaks. Besides, the magnitude of both peaks reduces. Secondly, a new VDOS appears for Ge beyond its Debye frequency (between 10 and 13 THz). For $L_{p}=1.1$ nm, two small peaks around 10.5 THz and 12 THz appear in the VDOS of Ge, matching the peaks of Si at those frequencies. The magnitude of the peaks for Si around the two frequencies is also enhanced with the reduction of period length, especially for the peak around 12 THz. The newly generated peaks in Ge and enhanced peak magnitude in Si around 10.5 and 12 THz indicate the formation of new modes in SiGe SLs with short period lengths, which is a strong signature of coherent phonon transport. In general, the reduction of SL period length increases the VDOS overlap of Si and Ge, resulting in a significantly reduced atomic vibration mismatch between Si and Ge atoms. As a result, the SiGe SLs with short period lengths behave more like uniform materials, which also promotes coherent phonon transport.

The coherent phonon transport can make phonons less scattered, thus increasing the TC. On the other hand, for SLs with large period lengths, the energy carried by Si modes beyond 10 THz can not transport through the Ge layers directly as no such modes exist in Ge. As a result, these phonons from Si have to convert to modes with lower frequencies. The frequent mode conversions at Si/Ge interfaces will reduce the energy transport efficiency, leading to a smaller TC. However, for SL with short period lengths, the mode between 10 and 13 THz in Si can travel directly through the Ge layers as those modes exist in both Si and Ge layers. Such a coherent transport eventually leads to the increase of TC with the reduction of period length. Furthermore, when the core-shell structure is introduced in the SiGe SL NWs, the magnitude of VDOS peaks around 9 THz in Ge and around 15 THz in Si is slightly reduced. In contrast, the VDOS peaks around 12 THz are slightly broadened for both Si and Ge, which means more new modes belonging to CSSL NWs are generated. As a result, there will be more modes in Si that can travel directly through the Ge layers, reducing the energy conversion probability at the interfaces. Such an effect enhances the coherent phonon transport and promotes heat transfer in CSSLs.

To further characterize the coherent transport, we calculated the mode weight factor for Si and Ge based on lattice dynamics calculations. The advantage of using the mode weight factor to characterize the coherent transport is that it can reveal the energy distributions of a specific mode on different regions or atom types. Thus it allows us to quantify how Si and Ge atoms share the vibrational energy of all modes. The mode weight factor of Si or Ge atoms for a specific mode λ can be defined as [37,38]:(2)$$f_{J,\lambda }=\sum_{i,\alpha }{\left(u_{i\alpha ,\lambda }\right)}^{2}$$
where $u_{i\alpha ,\lambda }$ denotes the eigenvector of atom *i* in direction α (x, y or z) for mode λ. The summation of atom index *i* accounts for all Si or Ge atoms. Here, we have $f_{Si,\lambda }+f_{Ge,\lambda }=1$. Figure 4 illustrates the calculated mode weight factor of Si and Ge atoms in SiGe SL NWs and CSSL NWs. At low frequencies (ω<4 THz), most of the mode energy is distributed on Ge atoms due to its larger atomic mass compared to Si. This is because in the acoustic-like mode, atoms with larger mass vibrate with larger amplitudes. For SL NWs with a larger period ($L_{p}=4.4$ nm, corresponding to the transition period length from incoherent to coherent phonon transport), only a small portion of the mode energy is distributed on Ge atoms beyond 10 THz (the Ge Debye frequency). This phenomenon is normal, as only a small portion of Ge atoms participate in those modes with $L_{p}=4.4$ nm according to the VDOS. One can imagine that the distribution of mode energy on Ge atoms beyond 10 THz will become less and less with the increase of the SL period. When the period length is decreased to $L_{p}=1.1$ nm, the mode energy distribution becomes more balanced in the frequency intervals of [5, 7.5] THz and [10, 16] THz, especially in the range beyond 10 THz. The increased mode weight factor of Ge atoms beyond 10 THz indicates that more and more Ge atoms participate in those modes with the decrease of $L_{p}$. As a result, the atomic vibrations between Si and Ge atoms become more coherent, which is in agreement with the VDOS analysis. With the introduction of core-shell structures in each SL layer, the mode weight factor of Ge atoms beyond 10 THz further increases, indicating that Ge atoms will share more energy with Si atoms for those modes. Hence, the newly generated interfaces by the core-shell structure can further enhance the coherent phonon transport.

Since more interfaces can reduce the vibration mismatch between different atom types, thus enhancing the coherent transport, we designed a phononic SL NWs with alternative Si and Ge blocks in each stacking layer of SLs (insert of Figure 5). We name this structure phononic SL NW. By changing the Si and Ge block sizes in each SL layer, we can change the interface density and thus may tune the coherent phonon transport strength.

Figure 5 depicts the TC of phononic SL NW as a function of Si or Ge block size *W* with $L_{p}=1.1$ nm. $W=L_{w}$ represents a normal SL NW. With the reduction of block size *W*, the TC of phononic SL increases noticeably, which is consistent with our considerations. TC is large for small W, which we attribute to increased surface contact between Si and Ge domains and stronger coupling of vibrational modes. When the phononic blocks become smaller, the entire structure becomes more homogeneous, and more new phonons distinct from those in Si and Ge NWs are generated. Those phonons can travel a long distance without getting scattered, thus enhancing thermal transport.

To illustrate the newly formed modes and the mode energy distributions, we also analyzed the VDOS and mode weight factor in phononic SL NWs with $L_{p}=1.1$ nm. Figure 6a shows the VDOS with different phononic block sizes. With the decrease of phononic block size *W*, the VDOS of Ge around 9.5 THz and Si around 15 THz reduces. Such behavior is similar to that observed in SL NWs when the period length is shortened. However, the VDOS of both Si and Ge between 10 and 14 THz is enhanced, and new VDOS peaks appear in this frequency range. When the phononic block size *W* is reduced to 0.55 nm, the VDOS peaks around 12 THz for both Si and Ge are split into two peaks. Besides, new peaks are formed at 14 THz for both Si and Ge. The VDOS curves in phononic SL NWs with $L_{p}=1.1$ nm and W=1.1 nm are very different from the corresponding components in NW state, indicating these newly formed modes belong to phononic SL. Consequently, they can travel through both Si and Ge regions without being scattered by the interfaces. One can also find that although the magnitudes of VDOS for Si and Ge are different, the shape of their VDOS between 10 and 14 THz is very similar, indicating that the VDOS of Si and Ge in this frequency range originates from the same phonon modes. The increased energy distribution on Ge atoms beyond 10 THz revealed by the mode weight factor further indicates that the vibrations of Si and Ge atoms become more balanced. The increased energy distribution on Ge atoms for the modes beyond 10 THz indicates that more Ge atoms participate in the vibrations of these modes with the decrease of phononic block size. Consequently, the coherent phonon transport is enhanced with more interfaces, which eventually facilitates thermal transport.

## 4. Conclusions

In conclusion, we performed equilibrium molecular dynamics simulations on the thermal transport of both SiGe CSSL and phononic SL NWs. Similar to SL NWs, the SiGe CSSL NWs possess a minimum TC at a period length of 4.4 nm. In CSSL NWs, the TC is further reduced as compared to that of SL NW beyond the period length with minimum TC (denoted as $L_{p}^{m}$) due to the enhanced scattering by the additional interfaces. However, below $L_{p}^{m}$, the TC of CSSL NWs is larger than that of SL NWs, which is ascribed to the enhanced coherent phonon transport. The enhanced TC has also been observed in phononic SL NWs, where numerous Si/Ge interfaces are generated. The maximum TC we obtained in phononic SL reaches up to 22.8 W/mK, which is 10 W/mK larger than that of SiGe SL NWs with the same period. The VDOS analysis demonstrates that new modes are generated between 10 and 14 THz with dense Si/Ge interfaces. The formation of new modes indicates that the entire structure can be treated as a homogeneous new material, which promotes coherent phonon transport. The mode weight factor also demonstrates that more and more energy is distributed on Ge atoms for the modes beyond 10 THz with the increase of Si/Ge interfaces, which facilitates the phonon transport through both Si and Ge regions.

## Figures and Tables

**Figure 1 nanomaterials-12-04373-f001:**
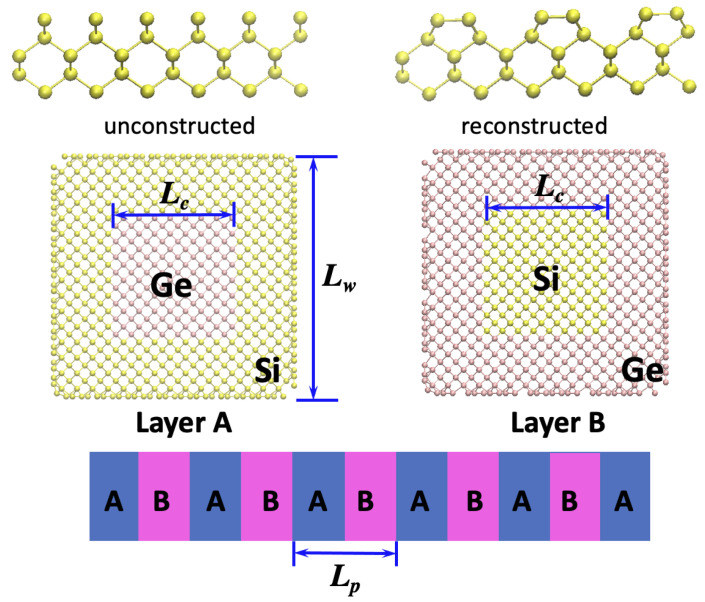
Schematic illustration of CSSL and the (2×1) surface reconstruction. Top: the atom arrangement for unreconstructed and (2×1) reconstructed {100} surfaces. Middle: the Si-Ge core-shell structures for the two primary stacking layers in CSSL (projected from the *z* direction.). Bottom: the CSSL NW formed by alternative stacking of A/B layers.

**Figure 2 nanomaterials-12-04373-f002:**
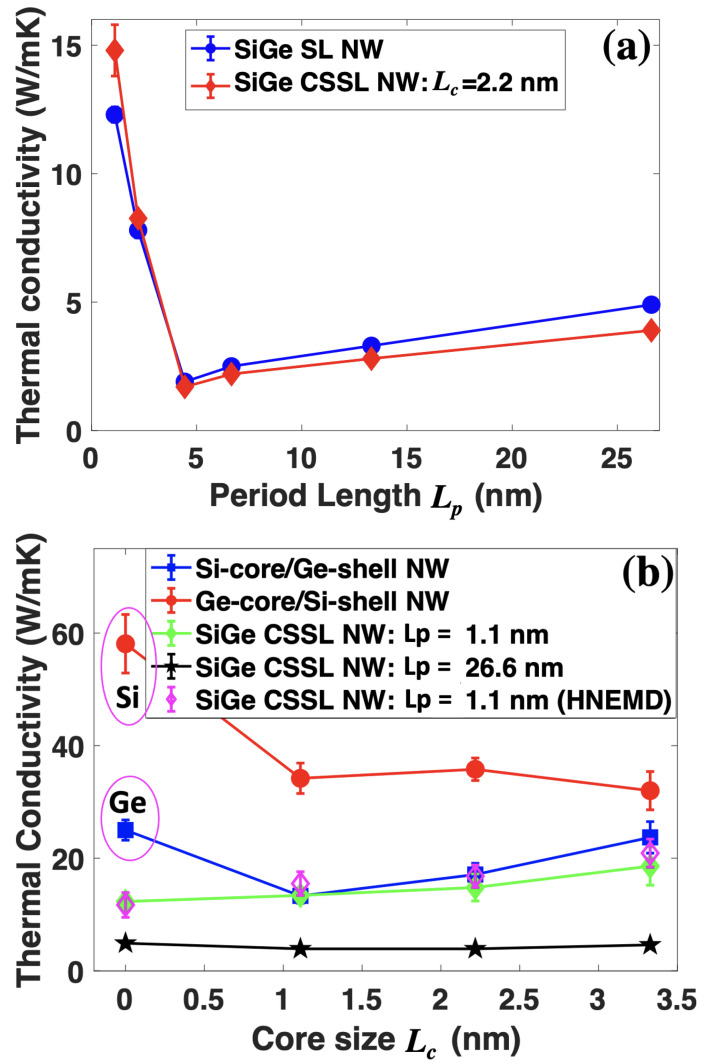
(**a**) TC as a function of period length $L_{p}$ for SiGe SL NWs and SiGe CSSL NWs with $L_{c}=2.2$ nm. (**b**) The variation of TC with the core size $L_{c}$ in SiGe core-shell NWs and CSSL NWs with different period lengths.

**Figure 3 nanomaterials-12-04373-f003:**
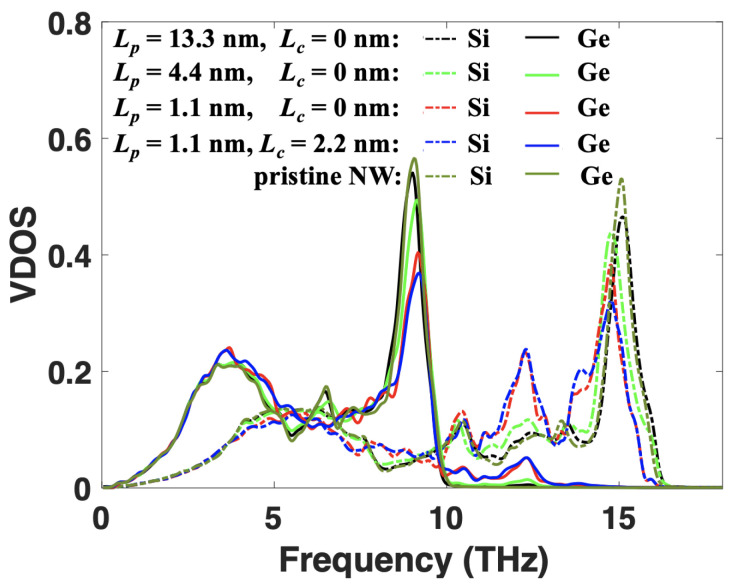
VDOS values of Si and Ge atoms in SiGe SL NWs ($L_{c}=0$) with indicated period lengths and SiGe CSSL NWs with specified period length and core size. The pristine NW denotes pure Si or Ge NW.

**Figure 4 nanomaterials-12-04373-f004:**
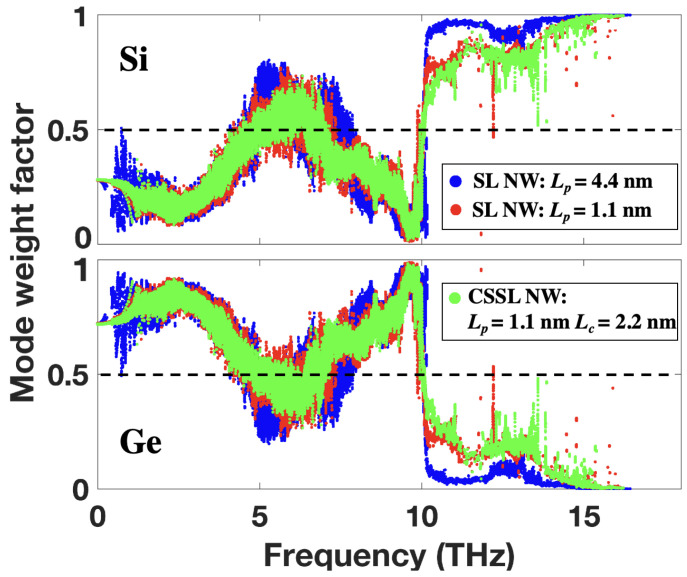
Mode weight factor of Si and Ge atoms in SiGe SL NW and CSSL NW with indicated period length and core size.

**Figure 5 nanomaterials-12-04373-f005:**
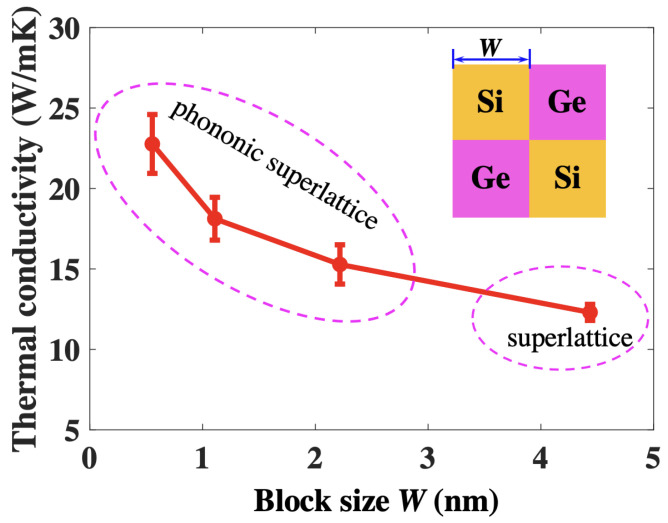
Thermal conductivity of phononic SLs varies with the phononic block size W with $L_{p}\;=\;1.1$ nm. The insert schematically illustrates the phononic blocks of a stacking layer in phononic SL NWs. The neighboring layers have opposite Si and Ge stacking orders.

**Figure 6 nanomaterials-12-04373-f006:**
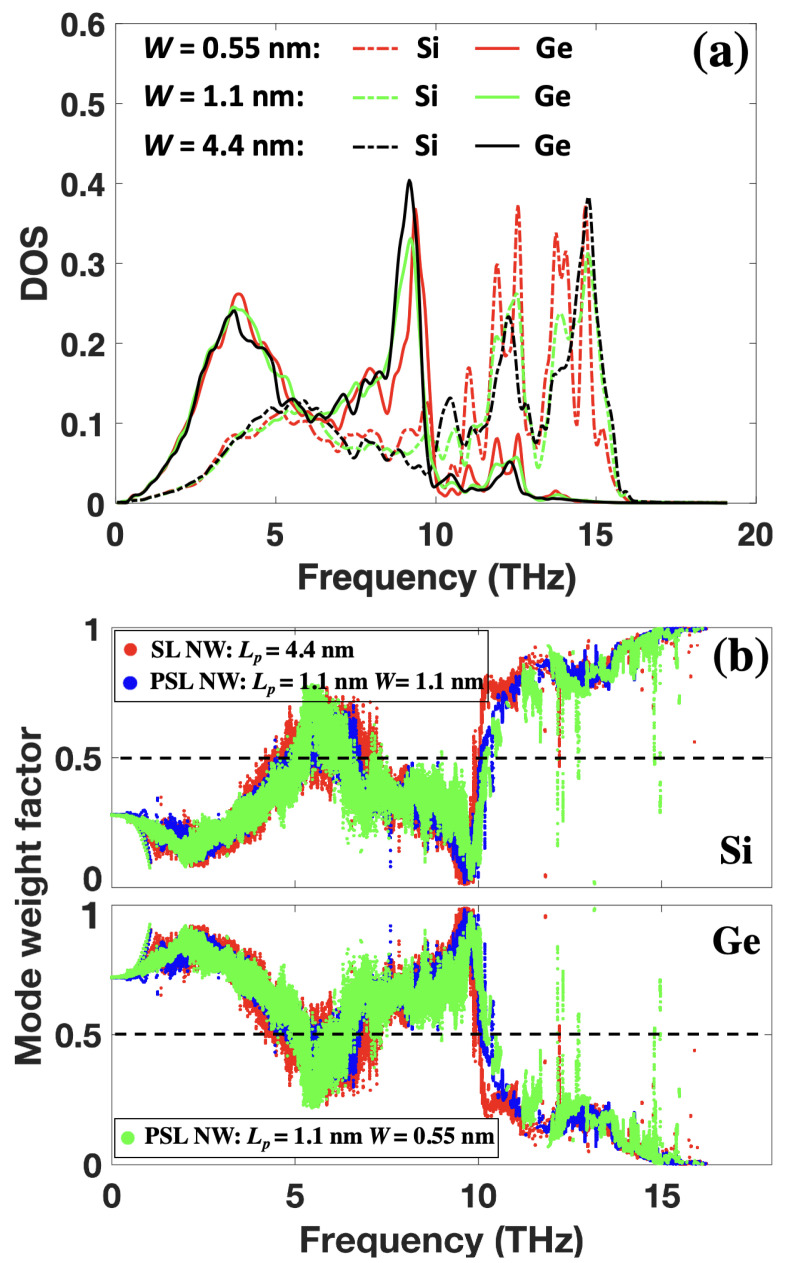
VDOS (**a**) and mode weight factor (**b**) of Si and Ge atoms in phononic SL NWs.

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
