# Peer review of "Enhancing the Coherent Phonon Transport in SiGe Nanowires with Dense Si/Ge Interfaces"

_nanomaterials, 2022, doi:10.3390/nano12244373_

Round 1

Reviewer 1 Report

This is solid piece of theoretical work where the authors addresses SiGe nanowire superlattices as nanoscale systems for engineering  the coherent phonon scattering as well the thermal conductivity (k)

Radial and axial interfaces are taken into account, macroscopic (k) and microscopic (DOS, mode weight factor) parameters are calculated for different nanostructures. The energy range 0-20 THz is considered in the calculations.

The authors should increase the discussion of literature and make an effort to highlight the novelty of the methods (if any) and of the results.  

Author Response

Thank you for your useful comments and insightful suggestions, which have helped us improve our manuscript. We have replied to your comments point by point, and the corresponding revisions were made in the manuscript. All changes have been highlighted in the updated manuscript.

Comments: This is solid piece of theoretical work where the authors addresses SiGe nanowire superlattices as nanoscale systems for engineering  the coherent phonon scattering as well the thermal conductivity (k)

Radial and axial interfaces are taken into account, macroscopic (k) and microscopic (DOS, mode weight factor) parameters are calculated for different nanostructures. The energy range 0-20 THz is considered in the calculations.

Answer: We thank the referee for the favor of our work.

Question: The authors should increase the discussion of literature and make an effort to highlight the novelty of the methods (if any) and of the results.

Answer: In the updated manuscript, we have added more literature review in the introduction part. Moreover, detailed discussions on the methods and on our obtained results are added in the revised version of the manuscript.

Reviewer 2 Report

The paper presents up-to-date results concerning thermal transport in SiGe superlattice (SL) nanowires. Authors found that depending on superlattice period lengths, the thermal conductivity can be reduced by introducing additional Si/Ge interfaces or enhanced due to the increased interface density for a small value of superlattice period lengths. In my opinion, this is an interesting result and the paper is worth publishing. However, prior to acceptance, the authors should consider and respond to the remarks below.

1) The Green-Kubo (GK) method was derived for the bulk limit [R. Kubo, J. Phys. Soc. Jpn. 12, 570 (1957); R. Zwanzig, Annu. Rev. Phys. Chem. 16 (1965)]. So, the Green-Kubo formula in such systems as nanowires is used as an assumption that needs to be verified by testing it on a case-by-case basis. Taking this into account and also noticing that the studied nanowires are inhomogeneous and have a lot of Si/Ge interfaces two questions arise:
a) about the convergence of the GK method in the strong confinement and inhomogeneous system as SLs with core-shell and phononic structures and lateral interfaces in SiGe SLs nanowires. It would be good to present convergence graphs.
b) about the correctness of obtained results. It would be good to compare the results obtained with the results obtained by another independent method - at least for a couple of points.

2) It will be good if the Authors compare of obtained results of thermal conductivity of pristine Si and Ge nanowires reported in the manuscript (or another size) with a reach literature data.

3) It should be also mentioned that the work [Limitations of Applicability of the Green-Kubo Approach for Calculating the Thermal Conductivity of a Confined Liquid in Computer Simulations, Computational Methods in Science and Technology 22, 197 (2016)] has shown that the GK method gives reliable results for the thermal conductivity along the nanowire and fails across nanowire. Did the authors observe the same behavior?

4) Could the authors comment (or discuss) in the manuscript on the influence of the phonon confinement effect on the thermal conductivity of studied nanowires which has been discussed recently in the work [Role of the phonon confinement effect and boundary scattering in reducing the thermal conductivity of argon nanowire, J. Chem. Phys. 154, 054702 (2021)]? This effect has been confirmed experimentally [Direct observation of confined acoustic phonon polarization branches in free-standing semiconductor nanowires, Nature Communications volume 7, Article number: 13400 (2016)].

5) The work lacks detailed information on the studied systems, such as the size of all the studied systems (number of particles, sizes of supercells, etc.). The data should be sufficient for other researchers who might wish to replicate or perform other studies of these systems.
Besides, the details of the simulations should be supplemented with the following information:
- which thermostat was used,
- which method of integration of equations of motion was used,
- what correlation times in the Green-Kubo method were used?

6) A considerable number of abbreviations makes reading the manuscript difficult.

7) Perhaps the authors would like to enrich the introduction with basic and more recent works on heat transport in nanostructures:
a) Thermal conductivity of individual silicon nanowires, APPLIED PHYSICS LETTERS 83, 2934, (2003)
b) Theoretical phonon thermal conductivity of Si/Ge superlattice nanowires, JOURNAL OF APPLIED PHYSICS 95, 682, (2004)
c) Precise control of thermal conductivity at the nanoscale through individual phonon-scattering barriers, NATURE MATERIALS 9, 491, (2010)
d) Coherent Thermal Conduction in Silicon Nanowires with Periodic Wings, NANOMATERIALS 9, 142, (2019)
e) Silicon Nanowires: A Breakthrough for Thermoelectric Applications, MATERIALS 14, 5305, (2021)
f) Ballistic Heat Transport in Nanocomposite: The Role of the Shape and Interconnection of Nanoinclusions, NANOMATERIALS 11, 1982, (2021)
j) Atomistic evidence of hydrodynamic heat transfer in nanowires, INTERNATIONAL JOURNAL OF HEAT AND MASS TRANSFER 194, 123003, (2022)
k) Effects of transverse geometry on the thermal conductivity of Si and Ge nanowires, SURFACES AND INTERFACES 30, 101834, (2022)

Author Response

Dear Referee,

please find the answers in the attached file, thank you!

kind regards,

Shiyun Xiong

Reviewer 3 Report

Dear Editor, I Reviewed the manuscript entitled “Enhancing the coherent phonon transport in SiGe nanowires with dense Si/Ge interfaces” by Cheng et al, finding it of interest. The work offers a better understanding of the coherent phonon transport, and studies the thermal transport in SiGe superlattice nanowires using equilibrium molecular dynamics simulations. However, I would like to outline some major concerns:

1-      In the “structures and methods” section the authors described the modeling and the theoretical approach. It is not clear what is the bond distance between the called A and B layers. The dimensions of Si and Ge are a little bit different between them, this means that not all bonds will have the same length. Since the phonon transport strictly depends on the atoms vibrational modes, hence the atomistic positions are important. The authors should report the bond lengths and remark the differences, if any, between the A and B layers.

2-      The authors described different simulation steps based on NPT, NVT and, finally, NVE ensembles, but they didn’t report the algorithm they used to control pressure, volume, and energy respectively. This information should be included in the “structures and methods” section.

3-      Figures 2 and 5 report the thermal conductivity of systems in function of Lp, Lc, and W. It is not clear if the reported values have been obtained at the end of simulations or if they represent average values obtained along all MD trajectories. The authors reported the estimated errors in figure 2a, then I suppose they adopted the second approach, but I don’t see details for the other data. Please add information about the reported results.

Author Response

(The authors gave the same response as above.)

Reviewer 4 Report

See attached file

Author Response

(The authors gave the same response as above.)

Reviewer 5 Report

In the paper "Enhancing the coherent phonon transport in SiGe nanowires with dense Si/Ge interfaces" based on equilibrium molecular dynamics simulations and lattice dynamics calculations, the thermal transport in SiGe superlattice nanowires with tuned Si/Ge interface density was investigated by using the core-shell and phononic structures as the primary stacking layers. It was found that the thermal conductivity decreased with the increase of superlattice period lengths (Lp) when Lp is larger than 4 nm.
There are a few comments on the paper:
1. Why was the (100) plane chosen for calculations? The model used has many drawbacks and cannot correspond to reality, since the artificially synthesized system is not relaxed and the sizes of silicon and germanium atoms are very different.
2. In the bulk state along the [001] direction in silicon and germanium, there is a helical axis 41, which is not traced in Fig. 1; clarifications are needed in the methodology on this issue.
3. In the description of Figure 3, it is written that the VDOS of Si and Ge atoms are close to the corresponding bulk values, but the values ​​themselves are not given. It also turns out that the artificiality of the system described in question 1 does not affect the VDOS of Si and Ge atoms, is it really so?

Author Response

Dear Referee,

thank you for your useful comments and questions! please find our answers in the attached file.

Best regards,

Shiyun Xiong

Round 2

Reviewer 2 Report

The work can be published after correcting inaccuracies in reference [31]:

Instead

"Hyzorek Krzysztof, T.K.V. Limitations of Applicability of the Green-Kubo Approach for Calculating the Thermal Conductivity of a Confined Liquid in Computer Simulations. CMST 2016, 22, 197–200."

should be

"Hyżorek, K.; Tretiakov, K.V. Limitations of Applicability of the Green-Kubo Approach for Calculating the Thermal Conductivity of a Confined Liquid in Computer Simulations. Computational Methods in Science and Technology 2016, 22, 197–200. https://doi.org/10.12921/cmst.2016.0000054"

Author Response

Thank you for your careful check. We have corrected the reference in the updated manuscript.

Reviewer 3 Report

The Authors considered all issue raised.

Author Response

Thank you very much for your support.